# Detection of Serum Allergen-Specific IgE in Atopic Dogs Tested in Northern Italy: Preliminary Study

**DOI:** 10.3390/ani11020358

**Published:** 2021-02-01

**Authors:** Morena Di Tommaso, Alessia Luciani, Paolo Emidio Crisi, Marica Beschi, Paolo Rosi, Francesca Rocconi, Arianna Miglio

**Affiliations:** 1Faculty of Veterinary Medicine, Veterinary University Hospital, University of Teramo, Piano d’Accio, 64100 Teramo, Italy; aluciani@unite.it (A.L.); frocconi@unite.it (F.R.); amiglio@unite.it (A.M.); 2Veterinary Clinic Dr. Paolo Rosi, via Brescia 46, 25086 Rezzato (BS), Italy; marica.beschi@gmail.com (M.B.); clinicaveterinariadott.rosi@gmail.com (P.R.)

**Keywords:** dog, atopic dermatitis, IgE, serology, allergy testing, Northern Italy

## Abstract

**Simple Summary:**

Canine atopic dermatitis (CAD) is a genetically predisposed allergic skin disease associated to IgE-mediated hypersensitivity. Serum Allergen-Specific IgE test (SAT) is used to detect allergen-specific IgE antibodies and to set-up a specific immunotherapy. The present study aims to identify the most relevant environmental allergens in a population of atopic dogs living in Northern Italy by using SAT, due to the absence of data in this area. Out of 117 selected client-owned dogs, 69 were included in the study. A screening test was used to identify indoor and outdoor allergens positivity and a specific serum allergy test was performed to detect an extended panel of allergens. Among the 49 positives to the screening test, 53% were positive for both indoor and outdoor, 38.8% only for indoor, and 8.2% only for outdoor allergens. Mites and pollen of *Rumex acetosa*, and grasses were the most represented allergens. This is the first study to collect data on the frequency of specific allergens involved in CAD in Italy by using SAT. In accordance with the result of this study, specific panels for geographical areas should be considered and re-evaluated at time intervals since numerous factors affect the prevalence of IgE positivity in atopic dogs.

**Abstract:**

Canine atopic dermatitis (CAD) is a pruritic allergic skin disease associated with IgE-mediated hypersensitivity. IgE is detected using Serum Allergen-Specific IgE test (SAT) in order to identify allergens. The present study aims to identify the environmental allergens in atopic dogs living in Northern Italy using SAT. The screening SAT (sSAT), using a monoclonal antibody cocktail-based ELISA to identify indoor and outdoor allergens, was performed. In all positive samples, an anti-IgE monoclonal antibody ELISA test was performed to extend panel of allergens. Out of 117 selected dogs, 69 were included in the study; 71% were positive and 29% were negative to sSAT. Among the 49 positive sSAT, 53% were positive for both indoor and outdoor, 38.8% only for indoor, and 8.2% only for outdoor allergens. This is the first study on the frequency of allergens involved in CAD in Italy using SAT. IgE hypersensitivity in atopic dogs of Northern Italy is usually associated with indoor allergens, primarily house dust mites. Among the outdoor allergens, an important role was played by *Rumex acetosa*. Polysensitization also commonly occurs. Therefore, since the numerous factors affect the IgE positivity in CAD, specific panels for geographical areas should be considered and re-evaluated at time intervals.

## 1. Introduction

Canine atopic dermatitis (CAD) is a genetically predisposed inflammatory and pruritic allergic common skin disease with characteristic clinical features associated with IgE-mediated hypersensitivity [1,2,3]. These antibodies are most commonly directed against environmental allergens, such as mite antigens, house dust, epidermal antigen, insect antigens, pollens, and mold spores [1,2,4,5]. Climate, vegetation, lifestyle, population density, and pollution levels are all factors that might contribute to the varying occurrence of CAD [6].

The diagnosis of CAD is based on history, clinical signs, and exclusion of other pruritic diseases [3]. The presence of allergen-specific IgE detected by Intradermal Test (IDT) or by Serum Allergen-Specific IgE test (SAT) is not useful for diagnosis due to the low specificity and sensitivity of these tests [7,8]. They do not have a good ability to discriminate between atopic and non-atopic patients [3]. In addition, in some dogs, clinical signs of CAD are present, even in the absence of detectable IgE (CAD-like) [4]. Only after clinical diagnosis of CAD is achieved, should IDT and/or SAT be used to identify the responsible allergen and to set up a specific immunotherapy [3,9]. Until recent years, IDT was the most commonly used method by dermatologists, however it is not of practical use for dogs and cats as patients need sedation and single inoculations of individual allergen extracts. In addition, numerous drug interferences have been demonstrated [10,11]. On the other hand, SAT is today widely used in clinical practice [10,11] due to the lack of risk and discomfort for the patient, the detection of multiple antigens by using a single serum sample, the quantitative nature of the results, the fact that the test can be performed on patients in the presence of a widespread cutaneous inflammation, and because it is less affected by of current drug therapies [12,13,14]. A serologic test is also required for large-scale investigations of CAD [15]. Since causative allergens vary according to geographical region, climate, environmental pollution/hygiene, and residential environments [15], knowledge of the geographic distribution of allergens is very important to set-up a panel targeted for a specific geographical area [16].

There are various studies from different countries around the world that investigate the prevalence of positive reactions to different aero-allergens in atopic dogs [6,15,17,18,19,20,21,22,23,24,25,26]. However, there are only a few studies on the distribution of allergens in Italian regions [27,28].

The present study aims to identify the most relevant environmental allergens in a population of atopic dogs living in Northern Italy by evaluating the seropositivity to allergen-specific IgE by using monoclonal antibody based enzyme-linked immunosorbent assay, due to the lack of data in this geographical area.

## 2. Materials and Methods

This study was carried out from May 2016 to October 2019, during different seasons and at the Veterinary private Clinic located in Brescia (Northern Italy). All animals tested were living in Northern Italy.

### 2.1. Criteria for Animal Selection

One hundred and seventeen client-owned dogs with clinical diagnosis of CAD were selected for the study. Diagnosis of CAD was made on the basis of history, the presence of characteristic clinical signs, i.e., at least 5 of 8 clinical features of CAD as described by Favrot et al. [29] (Table 1), and by excluding other skin condition with similar clinical signs and/or pruritic skin diseases that can resemble or overlap with CAD [3,7,29]. The administration of drugs such as glucocorticoids, oclacitinib, lokivetmab, antihistamines, and cyclosporine were discontinued at least 4 weeks before enrolment in the study.

### 2.2. Exclusion Criteria

Dogs with other pruritic skin diseases such as dermatitis due to *Malassezia*, superficial pyoderma, fungal skin infection, ectoparasitic, and neoplastic skin diseases were excluded from the study by performing appropriate diagnostic tests (skin cytology, multiple skin scrapings, hair plucking, hair combing, acetate tape impressions, ear swabbing, fungal culture) [7]. A strict elimination diet trial with commercial hydrolyzed protein was administered for at least 8 weeks prior to testing and an antiparasitic trial treatment was applied to rule out food allergy dermatitis and ectoparasite skin disease, respectively [7]. Subjects with CAD and concurrent conditions that needed treatment were excluded from the study.

### 2.3. Sample Collection and Analysis

Serum samples were collected and shipped on the same day to a reference laboratory (Univet Diagnostic Services, Barcelona, Spain) to perform SAT using monoclonal antibody cocktail-based enzyme-linked immunosorbent assay (macELISA, Greer Laboratories, Lenoir, NC, USA). This followed a previously well-established procedure [30,31] used to evaluate allergen-specific IgE in serum samples.

An anti-IgE macELISA screening serum allergy test (sSAT) was initially performed in order to identify the negativity or the positivity to significant levels of IgE against two categories: Indoor and outdoor allergens. The total number of allergens tested in each animal was 25 (10 indoor and 15 outdoor allergens), as is commonly used in European laboratories. The composite of the indoor allergen panel (IN) consisted of a mixture of 2 house dust mites, 3 storage mites, 3 molds, *Malassezia*, and flea antigens. The composite of the outdoor allergen panel (OUT) consisted of a mixture of pollen from 5 grasses, 6 weeds, and 4 common tree allergens. All results were expressed as ELISA Absorbance Units (EAU) and a cut-off of 150 EAU was established [30,31].

Subsequently, to identify the composition of allergen-specific immunotherapy, in all positive samples to IN and/or OUT an anti-IgE monoclonal antibody ELISA specific serum allergy test (UNITEST—ELISA, Greer Laboratories, Lenoir, NC, USA) was performed in the same laboratory. This detected an extended panel of allergens in the dogs:IN serum allergen-specific IgE tests: *Dermatophagoides farinae*, *Dermatophagoides pteronyssinus*, *Acarus siro*, *Tyrophagus putrescentiae*, *Lepidogliphus destructor*, *Alternaria alternata*, *Aspergillus fumigatus*, *Penicillium, Malassezia*, and fleas.OUT serum allergen-specific IgE tests: Phleum pretense, *Dactylis glomerata*, *Poa pratendis*, *Lolium perenne*, *Cynodon dactylon*, *Taraxacum vulgare*, *Artemisia vulgaris*, *Rumex acetosa*, *Plantago lanceolata*, *Parietaria* spp., *Chenopodium álbum*, *Platanus acerifolia*, *Olea europaea*, *Betula pendula*, and *Cupressus sempervirens*.

### 2.4. Statistical Analysis

Data analysis was performed using statistical software packages (MedCalc 17.9.2, MedCalc Software, Mariakerke, Belgium). All data were evaluated using a standard descriptive statistic. Normality was checked using the D’Agostino-Pearson test. A comparison between groups was done using the unpaired t test or the Mann–Whitney test, based on their distribution. For the statistical analyses, a *p* value < 0.05 was considered significant.

## 3. Results

Based on exclusion criteria, 48 dogs were rule out. Out of 69 atopic dogs included in the study, 44 (63.8%) were males and 25 (36.2%) were females. These subjects were aged between 2 and 10 years (median of 5 years). Various breeds were represented, such as Labrador Retriever (*n* = 11), French Bulldog (*n* = 10), German Shepherd dog (*n* = 8), American Staffordshire Terrier (*n* = 6), Mixed breed (*n* = 6), English Bulldog (*n* = 5), Rottweiler (*n* = 4), Bernese Mountain dog (*n* = 3), Shih Tzu (*n* = 2), Chihuahua (*n* = 2), and one subject for each of the following dog breeds: Yorkshire Terrier, Shar Pei, Weimaraner, Pug, Border Collie, Chow Chow, Dalmatian, Australian Shepherd dog, Akita Inu, Staffordshire Bull Terrier, American Pitt Bull Terrier, and Scottish Terrier.

Of the 69 screening tests performed, 49 were positive (71%) and 20 were negative (29%). Among the 49 positive screening tests, 26 were positive for both IN and OUT (53%), 19 were positive only for IN (38.8%), and 4 were positive only for OUT (8.2%). There were no significant differences between serum IgE levels of allergens in IN positive subjects compared to OUT positive subjects (*p* = 0.17).

All IN positive subjects (*n* = 45, 91.8%) showed a positivity for at least one allergen of the mite family, 12 for at least one allergen in the mold category, 11 for the *Malassezia*, and 1 for the flea saliva allergen. Serum IgE concentrations and distribution of various allergens in IN positive subjects is reported in Table 2 and Figure 1, respectively.

Frequent co-positives in the category of mites was found (Table 3). High co-positivity (>70%) was observed among *Dermatophagoides farinae* and *Acarus siro*, *Dermatophagoides farinae* and *Tyrophagus putrescentiae*, and *Tyrophagus putrescentiae* and *Acarus siro*.

All OUT positive subjects (*n* = 30, 61.2%) showed positivity for at least one allergen of pollen from grasses, 28 for at least one allergen of pollen from weeds and 21 for at least one allergen of pollen from trees.

Serum IgE concentrations and distribution of various allergens in OUT positive subjects is reported in Table 4 and Figure 2, respectively.

## 4. Discussion

Despite the widespread use of SAT in clinical practice, the detection of IgE levels for specific allergens involved in CAD using this method has not been studied in dogs from Italy. Only a few studies using IDT are reported in Italy [27,28], whereas there are various international studies from different geographical areas [6,12,15,32,33,34,35] that have investigated SAT in CAD.

Our study was performed in Northern Italy and 71% of the subjects tested were positive for a screening serum allergy test for indoor and/or outdoor allergens. The proportion of CAD cases positive to sSAT reported in different international studies ranged from 74.4% to 85.4% [6,21,36,37]. The actual prevalence of elevated IgE levels in dogs is not known [10,38]. The high proportion of positive results found in some studies is explained by the fact that the serum samples were submitted for IgE analysis only after clinical diagnosis of CAD was determined. In the present study, we found a lower rate of positivity compared with the aforementioned studies [6,21,36,37]. These results may be related to many factors such as the lower number of case studies, the type and number of allergen included in the serology panel, and the lack of performing IDT in addition to SAT to increase the diagnostic sensitivity [36,37], although there is no evidence that suggests that both need to be necessarily performed [7,10]. The percentage of CAD subjects positive to SAT reported in the present study is also lower than those found in the other studies conducted in Italy, although these studies are performed by using IDT [27,28]. These two testing methods are very different and a poor correlation of results between them is seen [7]; SAT measures circulating allergen-specific IgE and does not take into account other allergic pathways [2,39] while IDT is an indirect measure of cutaneous mast cell reactivity due to the presence of IgE [2].

Several assays for allergen-specific IgE serology testing, mostly based on ELISAs, have been used both in human and in veterinary medicine. These assays are performed to detect specific IgE antibodies against a panel of allergens (e.g., pollen, mold, house dust mite, and epidermal allergens) considered relevant for the patient. In evaluating serum-based testing, the International Task force on CAD reported that the methodologies for these tests varies from laboratories; few critical studies have evaluated the performance of these tests [10]. In the past decades, the detection of serum IgE was done using monoclonal, mixed monoclonal, or polyclonal anti-canine IgE. However, due to the higher sensitivity and specificity of a monoclonal antibody, the use of polyclonal anti-canine IgE antibodies has markedly decreased [10,40]. Good results have been demonstrated with the veterinary assay using a unique recombinant fragment of the extracellular portion of the human high affinity IgE receptor alpha-subunit (FcεRIα). It has shown a strong affinity for canine IgE other than a lack of cross-reactivity with the IgG [41,42]. Recently, the performance characteristics of a monoclonal antibody cocktail-based ELISA test as screening test for detection of allergen specific IgE in dogs (macELISA) has been thoroughly studied. High interlaboratory reproducibility and consistency of macELISA results have been documented between different laboratories [30,31]. Particularly, the variability between and among the European laboratories seem to be indistinguishable demonstrating that all laboratories are equally proficient in providing consistent results for all allergens tested. Moreover, the results from the macELISA are also directly comparable to those obtained with the assay using the alpha chain of human high affinity IgE receptor (Fc_R1; FcεELISA) [42]. The macELISA assay has also demonstrated to have an adequate uniformity of manufactured lots coated wells and of each new lot of allergen extract that is to be used in the well manufacturing process. For the aforementioned reasons, we decided to use the macELISA method since all the variables are well controlled and the laboratory that performed the analysis in our study was one of those included in the recent study that validated the intra and inter-laboratory reproducibility of the assay [31]. For the macELISA, a cut-off of 150 EAU has demonstrated to establish 99% confidence for positive responses, whereas the variability of this test is most prominent and of clinical relevance with low values that are around the cut-off point [31]. In our study, a few cases in the molds category resulted in being slightly below the cut off (between 100 and 150 EAU). This could question the negative test results obtained in the study.

Differentiating between positive and negative responses to sSAT is of fundamental importance for therapeutic purposes; it aims in selecting allergen extract to be included in an immunotherapeutic regimen [10,43,44]. It should be noted that these ELISA results are not quantitative but semiquantitative at best, and that the results are usually interpreted as qualitative only. There is no compelling evidence that the level of allergen specific IgE correlates with severity of clinical disease [10].

The results of serological IgE tests may vary markedly due to the geographical differences in exposure to different allergens and the lack of a universal standardization of allergens extracts. The appropriate selection of allergens to test is fundamental to obtain reliable SAT results. In fact, allergens, mainly pollens, are subject to a great geographic variability. Thus, it is important for veterinarians performing SAT to identify the allergens present in the regional location where the patients live. Information about relevant allergens can be obtained by consulting National Allergy Bureau (http://www.worldallergy.org/pollen/) [11]. It is of great clinical importance to update the allergen list based on climatic changes.

In our study, 29% of dogs tested showed a negative sSAT. These subjects might represent the CAD-like cases. In these cases, although patients present the same clinical signs of CAD, an IgE response to environmental or other allergens cannot be documented [4]. The real incidence of CAD-like is unknown [45]. In a retrospective study, Prelaud and Cochet-Faivre have obtained a CAD-like incidence percentage of 25.6%. In a more recent study, a percentage of 14,6% was found [36,37]. In dogs, the differentiation between CAD and CAD-like is usually made on the basis of the results of IDT or SAT [46]. Indeed, IDT and/or SAT support the definitive diagnosis identifying an IgE response [30]. As previously mentioned, the lack of performing IDT in addition to SAT, might explain the higher percentage obtained in the present study. Moreover, false negative results may also be explained by the involvement of untested allergens or due to low IgE levels that are below the cut-off point of test and therefore to a low sensitivity of the sSAT which is currently unknown.

In our study, we found that the concentrations of IgE of allergens in IN positive subjects showed no difference compared with OUT positive subjects.

Among the positive screening tests, 91.8% were positive for IN (includes both only IN and IN plus OUT) and 61.2% were positive for OUT (includes both only OUT and OUT plus IN). The high percentage of IN found in the present study is similar to another study conducted in Northern Europe [6]. These data can be explained by the high percentage of house dust mites positivity, which seems to represent the most important allergen involved in CAD in Europe [14].

Most of tested subjects in the present study are positive for more than one allergen and a high frequency (53%) for both indoor and outdoor allergens. This result overlaps with the studies previously carried out in Italy and in Europe and confirms that also in our area the phenomenon of polysensitization exists [6,12,24,27,28,33,46,47,48,49]. Interestingly, this was found both among the allergens present within the same panel, and particularly, mostly between house dust and storage mites, and between those present in the two different panels.

In the IN positive subjects, according to previous studies conducted in Italy and in other European countries, mites are the group of allergens that give the most positive results in allergological tests [17,19,23,27,33,50]. In temperate climates, CAD is predominantly associated with *Dermatophagoides* species [35,51]. Our data showed a high positivity in both storage mites and house dust mites groups. Particularly, *Dermatophagoides farinae*, *Tyrophagus putrescentiae,* and *Acarus siro* were the most represented mites. The percentage of positivity found for *Dermatophagoides farinae* (68.9%) was similar to those found in other Italian and European studies [14,19,23,27,28,33,50], whereas the sensitization rate for *Tyrophagus putrescentiae* and *Acarus siro* was higher than previous Italian investigations [27,28]. On the other hand, the sensitization to storage mites found in our study agree with that reported in other European studies [6,35]. In the present study, a high co-positivity was observed among these three mites (> 73.0%). Studies in dogs have revealed extensive cross-reactions between house dust and storage mites [46,52,53,54]. This high co-positivity rate could be due to cross-reactions related to the SAT method used [35] or to a greater predisposition towards storage mite allergens in the geographical area analyzed in the present study.

In our results, *Malassezia* and molds antigens have shown a percentage of positivity of 24,4% and 26,7%, respectively. It is well established that *Malassezia pachydermatis* is part of the normal cutaneous microbiota of dogs, although differing numbers of yeast occur at different anatomic sites. *Malassezia* antigens have been demonstrated to elicit a hypersensitivity response in atopic dogs [14,55]. In the literature, only few studies consider *Malassezia* as an allergen, thus not included in the majority of reports as they pre-dated its availability. In the international studies that use SAT to detect *Malassezia*, the positivity rate that was found widely varies and ranges between 0% and 60% [15,21,26,56]. It is interesting to note that in a recent study that evaluated both IDT and SAT, IDT showed a positivity for *Malassezia* of 24%, whereas no positivity was found to SAT [21]. In Italy, only Furiani et al. reported a percentage of positivity to this allergen of 35% by using IDT [27].

The importance of molds in atopic dogs is controversial. Previous reports demonstrate that sensitivity of the SAT to mold is lower than that of IDT, and SAT results are less reliable for detecting mold hypersensitivities [32,57]. Indeed, the positivity rate ranges between 0% and 44.8% [12,15,21,26,32]. In our case series, we found several values slightly below the cut off (between 100 and 150 EAU) in molds category. It seems that the IgE levels in this category are likely to be low, therefore if a lower cut off is used the number of positive tests would consequently increase. Nevertheless, the percentage reactions to mold allergens could also be explained by the variations in temperature and humidity, typical of some seasons or geographical areas.

Interestingly, in our study, only one subject was positive to antigen’s fleas. Likewise, other Italian and European studies have a medium-low incidence for this allergen [6,27,28,49]. In our study, these data can be justified as all examined subjects underwent a regular parasitic prophylaxis in their history and there was probably no sensitization to this allergen. However, since in the Greer antigen used for testing flea-specific IgE, is not extracted flea saliva but is crushed whole flea, negative reactions might simply result from a lack of sensitivity of the test.

In relation to positivity to outdoor allergens, it is difficult to make a comparison with other Italian and European studies, since, as previously stated, the distribution of pollen varies according to the geographical area and has varied over the years due to climate change. Regarding pollen, various factors such as geographical distributions and characteristics of plants, specific weather conditions, and season of the year must be considered.

In our study, data were collected throughout the years, thus comparing it with other Italian studies was difficult because either the studies tested different pollen or because a different method (IDT) was used. There was a high percentage of positivity to the OUT panel, which suggest that pollen may play a role in the development of CAD also in Italy. Particularly, pollen from grass showed a positivity of over 60% (*Cynodon dactylon* was the most represented). This was higher than those identified in other Italian studies where grasses showed a percentage of positive reactions < 30% by using IDT [27,28]. As expressed above, it is not possible to compare SAT and IDT methodologies. It is not known whether IDT is under-recognizing pollen allergies (low sensitivity) and/or SAT is reporting pollen allergies (low specificity) [21].

Weed groups also showed a high positivity (57.1%). Interestingly, *Rumex acetosa* (sorrel) was the only pollen of weeds that showed a high positivity (76.7%) in the present study. Sorrel is a weed that belongs to the family of the Polygonaceae and grows spontaneously. It is present in all regions of Italy, and blooms between May and August. There are no data on this allergen in other studies conducted in Italy. However, it is interesting to note that in a recent study conducted in Norway, a plant belonging to the genus Rumex (*Rumex acetosella*) was included in an allergy test and was the major positive allergen, with an incidence of 40% [6]. Similarly, sorrel was the only pollen allergen that appeared in the top four reactions in both the IDT and SAT tests performed in a study in South Australia [21]. In addition, Barili et al. reported a positivity SAT of sorrel of 38% [26]. In a recent Spanish study using SAT in horses with recurrent airway obstruction or atopic dermatitis, the most represented pollen allergens belonged to the genus Rumex [58]. The data relating to sensitization to this pollen obtained in the present study highlight the need to include this allergen in the outdoor panel also in Italy. Probably, the climate changes occurred over the last decade has allowed the spread of this pollen worldwide.

In this study, seropositivity to pollen of trees is found to be the lowest (42.9%) in the OUT, but high when compared to the positivity found in other Italian studies that used IDT [27,28]. Comparison with international studies is not possible due to testing different plants. In humans, cross-reactions between tree pollens are generally only significant between species of the same genera [21]. No data are available on this aspect in veterinary medicine. As previously mentioned, sensitivity and specificity of IDT and SAT for pollens are controversial.

Although this is a preliminary study and therefore it is necessary to increase the case series, the results of the present report confirm that the appropriate selection of allergens to test is fundamental to obtain reliable SAT results and that a continuous update of possible new environmental allergens is necessary, especially for those allergens, mainly pollens, subject to a great geographic variability. Thus, it is important for veterinarians who perform SAT to identify the allergens present in the regional location where the patients live; type and number of allergen included in the serology panel need to be continually adapted to the specific geographical area.

## 5. Conclusions

This is the first study showing data on frequency of allergens involved in CAD in Northern Italy by using a well-established monoclonal antibody-based ELISA test for detection of allergen-specific IgE in dogs.

This study shows that a high IgE hypersensitivity in atopic dogs of Northern Italy was most often associated with indoor allergens, primarily house dust mites, while mold and flea saliva play a marginal role. Among the outdoor allergens, an important role was shown by *Rumex acetosa*, which had previously never been considered in Italy. Polysensitization also commonly occurs.

In accordance to the results in this study, specific panels for geographical areas should be considered and re-evaluated at time intervals considering the numerous factors affecting the prevalence of IgE positivity in atopic dogs, and the continuous changes of climate conditions, pollination periods in individual areas, weather conditions, variations in distributions of individual plants, environmental hygiene, and residential environments.

## Figures and Tables

**Figure 1 animals-11-00358-f001:**
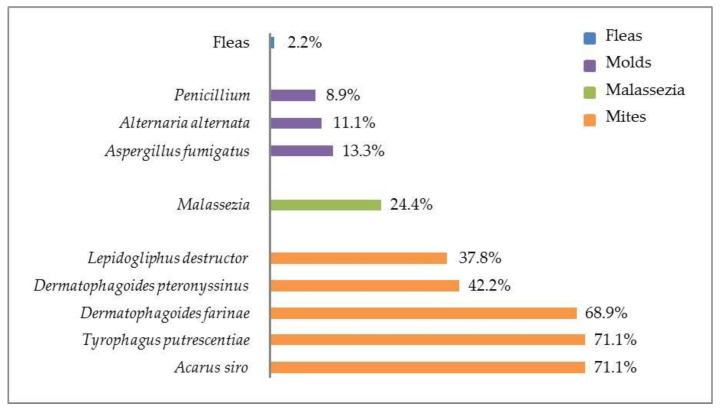
Distribution and percentage of various allergens in indoor allergen panel (fleas, molds, *Malassezia*, and mites) positive subjects (*n* = 45) with canine atopic dermatitis.

**Figure 2 animals-11-00358-f002:**
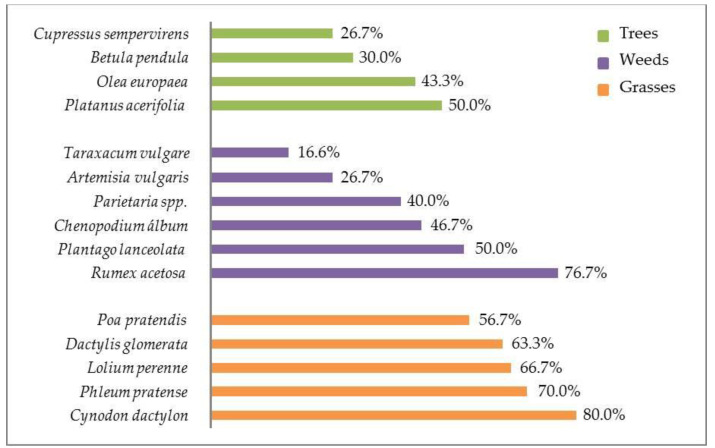
Distribution and percentage of various allergens in outdoor allergen panel (trees, weeds, and grasses) positive subjects (*n* = 30) with canine atopic dermatitis.

**Table 1 animals-11-00358-t001:** Favrot’s criteria. Set 1 (use for clinical studies).

No.	Criteria
1	Age at onset < 3 years
2	Mostly indoor
3	Corticosteroid-responsive pruritus
4	Chronic or recurrent yeast infections
5	Affected front feet
6	Affected ear pinnae
7	Non-affected ear margins
8	Non affected dorso-lumbar area

**Table 2 animals-11-00358-t002:** Serum IgE concentrations (ELISA Absorbance Units (EAU)) of various allergens in IN positive subjects (*n* = 45) with canine atopic dermatitis.

Allergens	No.	Median	IR
**House dust mites**			
*Dermatophagoides pteronyssinus*	19	304	240–455
*Dermatophagoides farinae*	31	935	373–2398
**Storage mites**			
*Lepidogliphus destructor*	17	335	210–562
*Tyrophagus putrescentiae*	32	678	353–1778
*Acarus siro*	32	640	270–2060
*Malassezia*	11	273	219–574
**Molds**			
*Penicillium*	4	223	168–332
*Alternaria alternata*	5	247	226–335
*Aspergillus fumigatus*	6	186	174–217
**Fleas**	1	3375	No IR

EAU, ELISA Absorbance Units; IN, indoor allergen panel; IR, interquartile range.

**Table 3 animals-11-00358-t003:** Number and (%) of co-positives in the category of mites in IN positive subjects (*n* = 45) with CAD.

Allergens	*Dermatophagoides farinae*	*Dermatophagoides pteron.*	*Acarus siro*	*Tyrophagus putrescentiae*	*Lepidogliphus destructor*
*Dermatophagoides farinae*		22 (48.9%)	34 (75.6%)	33 (73.3%)	19 (42.2%)
*Dermatophagoides pteron.*	22 (48.9%)		23 (51.1%)	22 (48.9%)	16 (35.6%)
*Acarus siro*	34 (75.6%)	23 (51.1%)		33 (73.3%)	16 (35.6%)
*Tyrophagus putrescentiae*	33 (73.3%)	22 (48.9%)	33 (73.3%)		18 (40.0%)
*Lepidogliphus destructor*	19 (42.2%)	16 (35.6%)	16 (35.6%)	18 (40.0%)	

IN, indoor allergen panel; CAD, canine atopic dermatitis.

**Table 4 animals-11-00358-t004:** Serum IgE concentrations (EAU) of various allergens in OUT positive subjects (*n* = 30) with canine atopic dermatitis.

Allergens	No.	Median	IR
**Grasses**			
*Poa pratendis*	17	448	335–1011
*Dactylis glomerata*	19	508	307–978
*Lolium perenne*	20	421	270–1086
*Phleum pretense*	21	387	214–811
*Cynodon dactylon*	24	494	293–1111
**Weeds**			
*Taraxacum vulgare*	5	827	327–3500
*Artemisia vulgaris*	8	479	252–1427
*Parietaria spp*.	12	277	211–776
*Chenopodium álbum*	14	310	205–414
*Plantago lanceolata*	15	435	297–552
*Rumex acetosa*	23	524	278–875
**Trees**			
*Cupressus sempervirens*	8	193	172–215
*Betula pendula*	9	277	197–441
*Olea europaea*	13	289	176–586
*Platanus acerifolia*	15	351	267–620

EAU, ELISA Absorbance Units; OUT, outdoor allergen panel; IR, interquartile range.

## Data Availability

The data presented in this study are available on request from the corresponding authors. The data are not publicly available due to privacy protection.

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
