# Peer review of "Detection of Serum Allergen-Specific IgE in Atopic Dogs Tested in Northern Italy: Preliminary Study"

_animals, 2021, doi:10.3390/ani11020358_

Round 1

Reviewer 1 Report

In the study the authors identify the most relevant environmental allergens in atopic dogs living in Northern Italy. A screening test by using Allergen-Specific IgE Serology (ASIS) was performed in these animals. There are very few studies about the frequency of allergens involved in CAD in Italy, and are carried out using IDT.  

The background is well described and references are complete. 

Methods, statistical analysis and results are complete and discussion is well-argued.

This is a preliminary study and the number of cases are limited, as the authors recognize. The paper doesn’t require revision.

Author Response

Dear Reviewer,

the authors are grateful for the positive comments and evaluation of the paper.

Reviewer 2 Report

The manuscript “Detection of Serum Allergen-Specific IgE in Atopic Dogs Tested in Northern Italy: Preliminary Study” by Di Tommaso et al., is an interesting research study to evaluate the frequency of allergens involved in Canine Atopic Dermatitis in Italy (the first study), due to the lack of data in this geographical area. In order to identify allergens, IgE is detected using allergen-Specific IgE Serology (ASIS). In particular, to perform ASIS the Authors used a well-established monoclonal antibody cocktail-based enzyme-linked immunosorbent assay (macELISA). The present study is very important for veterinary dermatologists who perform ASIS in North Italy to identify the allergens present in the location where the patients live and to set up a specific immunotherapy.

Few considerations:

Line 85: correct including the number of the reference: by Favrot et al. “[29]” (Table 1)….

  1. Favrot, C.; Steffan, J.; Seewald, W.; Picco, F. A Prospective Study on the Clinical Features of Chronic Canine Atopic Dermatitis 438 and Its Diagnosis. Vet Dermatol 2010, 21, 23–31, doi:10.1111/j.1365-3164.2009.00758.x.

In addition, according to journal’s policy, I think that ethical review and approval should be required for this study despite all the animals admitted to the Veterinary Clinic, with consent from the owners, were routinely undergo blood withdrawal and all the data were collected as part of the clinical investigations.

Author Response

Dear Reviewer,

the authors appreciate the suggestions.

Line 85: as suggested, we included the number of the reference

Ethical statement: for this study there is not ethical approval. However, all procedures were compliant with the European and national recommendations, and informed consent obtained from the owners also included the consent to participate in the study. These informations have been added to the manuscript.

Reviewer 3 Report

This paper makes clear the importance of a regional approach to allergy testing in order to provide the best therapeutic options for dogs with atopic dermatitis. 

I have very minor changes to recommend.

  1. In the Simple Summary, I would change "due to IgE-mediated" to "associated with IgE-mediated.  Atopic dermatitis is much more complex and this slight change is more compatible with what we know currently. You do use “associated with” in the Introduction and I think it is important to do so in the Summary as well.
  2. In the Simple Summary:  there seems to be an overemphasize on Rumex.   If you look at the data, 76.7% of the dogs had a positive test to this weed, however, over 50% of the dogs reacted to each of the grasses.   I would clarify by adding a phrase that Rumex and grasses were commonly positive.
  3. Dogs with Malassezia and pyoderma were excluded.  Given that these infections are often secondary to atopic disease, I am curious as to why these dogs were excluded.  Perhaps the percentage of Malassezia positive dogs would have been higher had they been included.  I would consider adding this reference to your discussion of Malassezia in the Discussion section:   Farver K et al.Humoral measurement of type-1 hypersensitivity reactions to a  commercial Malassezia allergen. Vet Dermatol 2005, 16:261-268.   In this paper they show with IDT a prevalence of 90+ percent positive when dogs have recurrent otitis and/or dermatititis.
  4. In the Methods section, I would like to see more information about the screening test.  It appears that if dogs were negative on the screening test, they were not analyzed further.   so the specificity, sensitivity, and predictive values of the test would be good to know.   e.g. How well does a negative screening test predict a negative ELISA?     How was the screening test validated and can you provide a reference for the screening test specifically?
  5. line 239.  I think it is always important to acknowledge that negative data are less reliable than positive data.   It is just as likely that the dogs with a negative screening test were negative due to allergen specific IgE levels below the detection limits of the assay OR the correct allergen was not tested.  This is alluded to at the end of the paragraph but lack of test sensitivity is not mentioned.
  6. Line 304.  I think it is important to acknowledge that since the antigen used for testing for flea-specific IgE is a whole body extract, negative reactions to fleas could simply result from a lack of sensitivity of the test.
  7. Paragraph starting with line 320. This paragraph is a little confusing to me. In figure it appears that the percentage of dogs with a positive reaction was 76.7% Yet in this paragraph you suggest it is 46.9%.   Can you clarify?
  8. I would recommend adding more detail to the figure legends for the figures.
  9. I would discourage you from creating new abbreviations that are not used that frequently.   I would delete ST for screening test, and just spell it out each time, and since ASIS is not commonly used, I would remove that as well. By contrast, ELISA is a well-established abbreviation and is worth keeping.

Author Response

Dear Reviewer,

the authors appreciate the suggestions.

1. and 2.: as suggested, we changed “due” to “associated” and we added grasses among the most represented allergens.

3.: dogs with Malassezia were excluded because they  required treatment (oral and/or topic drugs) in order to avoid potential influence on test results. We added the suggested reference.

4. and 5.: actually, the dogs negative on the screening test were not submitted to further analysis. Although the sensitivity is unknown, this screening test is an ELISA that uses a cocktail of the most of the same anti-IgE monoclonal antibody. Although there are not studies that provide robust data on the clinical usefulness of test for CAD, Lee et al. have demonstrated a high concordance of results comparing the macELISA and a high affinity IgE receptor-based ELISA. The lack of performance characteristics has added in a sentence on line 248 and these references are reported in the manuscript (Lee K.W et al., 2009 and 2012). 

6.: we added a sentence with this observation in the paper.

7.: sorry, this is a typo. We corrected the datum in the paper.

8.: as suggested, we added more informations to the figure legends and we changed Figure 1.

9.: since ELISA is also used to detect IgE receptor alpha-subunit, we changed ASIS (Allergen-Specific IgE Serology) with SAT (Serum Allergen-Specific IgE test), a more used acronym.